# Defining a Treatment Model for Self-Management of Fatigue in Rehabilitation of Acquired Brain Injury Using the Rehabilitation Treatment Specification System

**DOI:** 10.3390/jcm12093192

**Published:** 2023-04-28

**Authors:** Frederik Lehman Dornonville de la Cour, Anne Norup, Tonny Elmose Andersen, Trine Schow

**Affiliations:** 1Cervello, 2800 Kongens Lyngby, Denmark; 2Department of Psychology, University of Southern Denmark, 5230 Odense, Denmark; 3Neurorehabilitation Research and Knowledge Centre, Rigshospitalet, 2600 Glostrup, Denmark; anne.norup@regionh.dk

**Keywords:** neurological rehabilitation, fatigue, self-management, brain injuries, translational medical research, qualitative research

## Abstract

Systematic treatment descriptions to standardize and evaluate management of fatigue after acquired brain injury (ABI) are lacking. The purpose of this multi-phase qualitative study was to formulate a treatment model for promoting self-management of fatigue in rehabilitation of ABI based on practice-based understandings and routines. The study was conducted in a community-based rehabilitation center in Denmark. The model was defined using the Rehabilitation Treatment Specification System. Phase 1 comprised co-production workshops with five service providers (occupational therapists, physiotherapists, and a neuropsychologist) to elicit preliminary treatment theories. In Phase 2, four case studies were conducted on management of fatigue in vocational rehabilitation. Interviews (*n* = 8) and treatment log entries (*n* = 76) were analyzed thematically to specify treatment targets and active ingredients. The treatment model comprised five main components: (i) Knowledge and understanding of fatigue, (ii) Interoceptive attention of fatigue, (iii) Acceptance of fatigue, (iv) Activity management, and (v) Self-management of fatigue. For each component, lists of targets and active ingredients are outlined. In conclusion, management of fatigue includes multiple treatment components addressing skills, habits, and mental representations such as knowledge and attitudes. The model articulates treatment theories, which may guide clinical reasoning and facilitate future theory-driven evaluation research.

## 1. Introduction

Numerous people with acquired brain injury (ABI) struggle to manage persistent symptoms of fatigue [1], and effective treatment is needed to promote self-management in daily living [2,3]. Fatigue is common following ABI [4,5,6] and may contribute to disability [7], diminished quality of life [8], and lower rates of return-to-work [9,10]. With a lack of robust evidence for management of fatigue [11,12,13,14], current practice in neurorehabilitation includes a complex multi-component behavioral approach with the aim of empowering individuals to manage symptoms of fatigue independently [15,16]. In Denmark, this approach is commonly termed Energy Management (EM) and is widely used in clinical practice to reduce the burden of fatigue. It comprises various educational, psychological, and behavioral strategies such as psychoeducation, symptom monitoring, and planning of activities and rest. However, the specific content and the hypothesized active ingredients of treatment are poorly defined, and clinicians rely predominantly on their own knowledge and experience when managing fatigue [17,18,19]. Consequently, service provision may vary across sites and practitioners, and the lack of shared terminology and standard descriptions of treatment hinders evidence-based practice in fatigue management.

Inadequate descriptions of treatment are a long-standing issue in rehabilitation, frequently termed the “black box” problem [20]. The Rehabilitation Treatment Specification System (RTSS) [21] provides a framework for defining treatments in a systematic manner. In this framework, a “treatment component” comprises three elements: Ingredients (what the clinician does), Mechanisms of Action (how the treatment is expected to work), and Targets (aspects of functioning directly targeted for change). Targets are organized in three mutually exclusive groups: Organ Functions, Skills and Habits, and Representations (i.e., knowledge, attitudes, etc.). Defining a treatment includes determining (i) the number of treatment components, and for each component: (ii) The recipient, (iii) the target, and (iv) the ingredient(s) [21]. An extensive introduction to RTSS is available in The Manual for Rehabilitation Treatment Specification [22].

Using the RTSS as a heuristic tool, the aim of this qualitative study was to translate practice-based knowledge, understandings, and routines in fatigue management into a treatment model for promoting self-management of fatigue following ABI. The systematic descriptions of treatment provided by such a model may support decision-making in clinical practice and guide theory-driven research in building an evidence-base for management of fatigue.

## 2. Materials and Methods

A multi-phase qualitative study was conducted to specify a treatment model of EM based on investigations of clinical practice in management of fatigue. Phase 1 entailed co-production workshops with practitioners to formulate an initial draft of the treatment model. In phase 2, a collective case study was conducted to examine how treatment processes unfold in clinical practice. Based on case analysis, the initial draft was refined, finalized, and reported as the EM model. The treatment model was specified systematically based on RTSS using The Manual for Rehabilitation Treatment Specification [22]. The model was developed in the context of vocational rehabilitation, and the target population of the treatment model was defined as adults with ABI reporting persistent and debilitating fatigue (≥3 months post-injury). Formal assessment and other treatments not concerned with management of fatigue were outside the scope of the model.

The study was carried out from 2018–2022 in a community setting at a specialized brain injury rehabilitation center in Denmark. The Danish health care system is tax-based, and rehabilitation is provided free of charge for the service user. The study was conducted in accordance with the Declaration of Helsinki [23] and reported using Standards for Reporting Qualitative Research [24]. By Danish regulation, ethics approval was not required by the Danish National Ethical Committee System due to the study design.

### 2.1. Phase 1: Co-Production Workshops

Initial treatment theories were specified in a co-production process with five experienced clinicians from 2019–2020, including occupational therapists, physiotherapists, and a neuropsychologist (characteristics are provided in Appendix A). A series of exploratory workshops were conducted in an iterative process to elicit a preliminary understanding of treatment components. First, three mono-disciplinary workshops were conducted followed by four interdisciplinary workshops with all group members. Researchers facilitated discussions and produced drafts of an initial treatment model for review at the following meeting. At the end, draft materials were finalized, revised, and approved by the group. Initial treatment theories included detailed ingredients related to, e.g., knowledge, understanding, and acceptance of fatigue, and planning, prioritizing, and adapting activities. The results of this preliminary work in phase 1 were incorporated into the final EM model.

### 2.2. Phase 2: Collective Case Study

All individuals referred to vocational rehabilitation were screened for eligibility consecutively over 13 months in 2020–2021. Inclusion criteria for the case study were: (1) 18–65 years old, (2) ABI, (3) 3–24 months post-injury, and (4) clinically significant fatigue, using the Lynch et al. [25,26] case definition (assessed with a structured interview). Participants were excluded in case of (1) neurodegenerative disease or progressive central nervous system disease (e.g., tumor or multiple sclerosis), (2) referral of concussion/mild traumatic brain injury, (3) any comorbidity causing fatigue, (4) active alcohol or substance abuse within the latest three months, (5) retired or withdrawn from the labor market, (6) severe cognitive or communicative deficits interfering with participation, or (7) severe mental distress to a degree in which study participation interferes with rehabilitation. Participants provided written informed consent, and researchers had no relationship with service users prior to study commencement. In Denmark, 6102 adults between 18–64 years old are hospitalized with an ABI per year [27]. However, it is unclear how many of these need vocational rehabilitation after hospital discharge.

#### 2.2.1. Qualitative Data

For each case, two interviews were conducted by author F.D. one-on-one with the service user (ca. 45–75 min.) and a service provider (ca. 45–60 min), respectively, at the end of rehabilitation using a realist approach [28]. Interview guides were developed using resources from the RAMESES II project [29] for conducting realist interviews to investigate treatment theory. Service users were interviewed at home, and service providers were interviewed in online video meetings. Field notes were made during interviews, and interviews were audio-recorded and transcribed verbatim. Further, a RTSS-based treatment log was administered to service providers. Following each treatment session, service providers made a log entry comprising a summary of up to three pairs of ingredients, mechanisms of action, and targets. The log was pilot tested and revised prior to the case study, and service providers received instructions to the log and an illustrative example.

Interview transcripts (*n* = 8) and log entries (*n* = 76) were analyzed thematically. A coding protocol was developed based on a realist approach [30] using RTSS terms to organize codes on a first-order level (categories are defined in Table 1). On a second-order level (within each category), data units were coded de novo to describe salient features of the data units. Coding was conducted by F.D. and supervised, reviewed, and verified by T.S. Next, F.D. sorted codes into cardinal themes and sub-themes by comparing codes with each other in an iterative process. Finally, relationships among themes in the Ingredients and Targets categories were identified and organized as treatment components. Qualitative analysis was conducted using NVivo 1.6.1 [31].

#### 2.2.2. Descriptive Data

A survey was administered to service users once a week during rehabilitation to monitor day-to-day energy levels (on a 0–10 scale) and working h/wk. The survey was pilot tested on the first case. Additionally, service users completed six standardized paper-based questionnaires at inclusion and end of rehabilitation. Questionnaires included measures of: fatigue, the Dutch Multifactor Fatigue Scale [32,33]; sleep quality, the Pittsburgh Sleep Quality Index [34]; affective symptoms, the 21-item Depression Anxiety Stress Scales [35]; health-related quality of life, the 5-level EQ-5D [36,37]; self-efficacy, the General Self-Efficacy scale [38]; and work readiness, the Readiness for Return-To-Work Scale [39]. Quantitative data were analyzed using graphs and descriptive statistics in R version 4.2.0 (R Foundation for Statistical Computing, Vienna, Austria) [40] using the ggplot2 package [41].

### 2.3. Model Specification

The treatment components identified in case analysis in phase 2 were compared with the components of the initial draft of the model derived from phase 1. The final organization of treatment components and the targets and ingredients of each component were respecified based on results from both phases using The Manual for Rehabilitation Treatment Specification [22]. Targets and ingredients identified in both phases were merged and incorporated into the final treatment model, the EM model.

## 3. Results

Five out of ten screened were enrolled in the case study. Three were not eligible (no fatigue), and two declined due to lack of energy. One participant was withdrawn after inclusion due to severe mental distress. Thus, four participants completed the study. Characteristics are provided in Table 2. The number of treatment sessions ranged from 11 to 28, and the duration of rehabilitation ranged from 23 to 45 weeks (some programs were delayed by lockdowns due to COVID-19). Timelines of the rehabilitation programs, including self-reported day-to-day energy levels, are illustrated in Figure 1. Detailed case descriptions are provided in Appendix B, and outcomes on standardized questionnaires are reported in Appendix A.

### 3.1. Treatment Model on EM

Five treatment components were specified in the EM model. Targets and ingredients of these components are listed in Table 3.

#### 3.1.1. Knowledge and Understanding of Fatigue

This component was concerned with enhancing knowledge and understanding of fatigue as a sequelae of ABI, including the nature of ABI-related fatigue, triggers of fatigue symptoms (e.g., mental vs. physical exertion), limitations posed by fatigue and needs due to fatigue (e.g., breaks and pacing of activities), and how symptom fluctuations interact with activities. Common ingredients were provision of general information about fatigue and discussion of information in relation to individual experiences:

“She (neuropsychologist) can explain things, I cannot explain to myself, because I do not understand what is going on. She can explain a bit more technically, why certain things happen (…).”(Case B, service user)

Fatigue/activity diaries were provided to examine patterns in the interaction of energy levels and activities, and clinicians guided interpretations of diary entries and probed recent experiences (to improve awareness of straining/relieving activities). Challenges faced during work trials were used to discuss difficulties and needs due to fatigue. Educational material such as visual models was used to teach about consequences of fatigue.

#### 3.1.2. Interoceptive Attention of Fatigue

Interoceptive attention of fatigue refers to the ability to notice, interpret, and respond to signs of fatigue. The target was to form a habit of attending to early signs of fatigue in daily life:

“I have become more aware of what kind of small signs I need to pay attention to. Like, I can feel my concentration starts failing a bit, I need to spend more energy to focus on something, or I simply start to yawn.”(Case A, service user)

Ingredients for improving interoceptive attention included provision of a fatigue/activity diary (to direct attention to fatigue/energy levels) and a 10-point energy scale (to delineate sensations of fatigue/energy). Recent experiences in everyday life and challenges faced in work trials were queried to elicit reflections upon signs of fatigue. Mindfulness was practiced as a technique to guide attention to bodily sensations. In the case of overt signs, cues were provided. When noticing worsening of symptoms, service users were encouraged to redirect behavior to avoid overexertion, e.g., interrupting the current activity, taking a break, and resisting any impulse to push.

#### 3.1.3. Acceptance of Fatigue

Acceptance of fatigue is concerned with changing attitudes regarding acknowledgement of fatigue and expectations of oneself in relation to the current functional level. Participants described increased recognition of fatigue as a chronic condition interfering with daily life:

“I have realized that I need some breaks and rests, and I need to do some things differently than before. I am more realistic with myself. So, I have also set more realistic goals for myself.”(Case D, service user)

Clinicians were involved in the acceptance process by discussing fatigue and expectations for recovery/persistence of fatigue. One service provider described using formal assessment as a means to foster recognition (i.e., validating experiences of fatigue as a legitimate problem by documenting fatigue).

#### 3.1.4. Activity Management

Activity management concerns the acquisition of skills and formation of habits for managing fatigue in daily life. This target concerns forming a habit of (i) scheduling and (ii) planning daily activities with consideration of fatigue, (iii) alternating periods of activity and rest, (iv) reappraising activities, and (v) managing tasks efficiently. It also concerns (vi) improving the ability to rest and relax.

Scheduling daily activities was encouraged using, e.g., week planners or daily schedules. Clinicians encouraged regular rests, distributing exerting activities throughout the day and week, planning rest ahead of exerting activities (e.g., work), and making room in advance for recovery after expected exertion:

“(…) and when planning a week, well, then I can say already: This week (…), That gets tough. I know that, but then I know: Well, then I am not going to do anything at all that day. Right, I have to stop there and say: Well, I need to sleep there, and then I can do a little there.”(Case C, service user)

Clinicians queried preferences for resting, suggested relaxation techniques (e.g., mindfulness, listening to music, walking, etc.), and directed service users to test and practice strategies. Further, clinicians encouraged intermittent breaks and reminded subjects to rest throughout the day. Reappraising activities refers to a habit of prioritizing valued activities, reconsidering how one manages activities of daily living (redelegating or restructuring tasks and duties), and being better at “saying no” to others to take care of oneself (e.g., turning down invitations or cancelling appointments):

“We talked a lot about drag of fatigue, for example. When he (service user) happened to push himself (…) What it did to him, and what price he paid to… You know, by overloading himself. Then he… Well, we talked a lot about that it was a matter of prioritization. Sometimes, it is okay to get fatigued, but one needs to have planned for… To be able to get a good rest afterwards.”(Case B, service provider)

Finally, a theme was about improving efficiency of task management, e.g., breaking down tasks, planning ahead for task engagement, slowing down to avoid mistakes, and alternating between tasks.

#### 3.1.5. Self-Management of Fatigue

Self-management of fatigue refers to feeling confident knowing how and when to use strategies independently in different situations to prevent symptoms of fatigue from worsening and to manage symptoms, when they do get worse. Thus, this target is concerned with changing beliefs or attitudes about one’s ability to manage fatigue in daily living, i.e., self-efficacy in relation to fatigue management:

“Well, I have learnt to get better at noticing my fatigue, and then I also know how to react when I start to notice that feeling of fatigue. Like, now I need a break, now I need to sit down, or now I need to lie down.”(Case A, service user)

Clinicians queried the options available if fatigue gets worse in different situations (to boost a sense of autonomy).

## 4. Discussion

Rehabilitation services for fatigue in adults with ABI encompass complex multi-component treatments without sufficient evidence or standard descriptions. This study defines a model outlining hypotheses on how a range of actions in treatment are expected to affect aspects of functioning related to self-management of fatigue. Previous work has provided guiding principles and strategies for fatigue management in the scope of larger rehabilitation programs [16], and ongoing research aims at developing a management program for post-stroke fatigue based on perspectives of clinicians and stroke survivors [42]. Using RTSS, this study facilitates theory-driven evaluation research by formulating testable hypotheses of individual treatment components based on current practice. The EM model may contribute to bridge the gap between research and practice by guiding future research, which may refine components of the model and inform clinical practice.

This model directs treatment and provides guidance in treatment planning. However, it is not intended to script or manualize practice in a fixed program. Treatment components may be delivered in any order or simultaneously with one another depending on the targets of treatment and the hypothesized mechanisms of action. Person-centered rehabilitation is widely recommended for clinical practice [43] and given the complex interplay of multiple factors contributing to the impact of fatigue—and the vast heterogeneity of individuals with ABI—rehabilitation services need to be tailored to the unique characteristics and circumstances of the individual. Providing tailored services is complex and challenging, and standard descriptions of treatment may guide clinical reasoning and decision-making in planning and provision of individualized treatment [44].

The aspects of functioning targeted by EM may be causally related to one another. For example, better understanding of fatigue may promote acceptance, which in turn facilitates management of fatigue in daily life, as proposed in previous research [1,45,46]. Targets may also affect distal clinical outcomes such as mood, daily life functioning, and vocational capability. Such relationships are outside the scope of the treatment model, however, a better understanding of these is integral to determine the clinical utility of EM. Future work on this matter may advance theories concerning self-management of fatigue and inform clinical decisions on the timing of treatment components, interpretations of outcome patterns, and concerns regarding who will benefit clinically from which treatment component [47].

As fatigue is a complex and multifaceted problem, a promising strategy for improving clinical outcomes is to screen and treat potentially modifiable contributors to fatigue [48,49]. For example, Case B complained of poor sleep, abnormal circadian rhythm, and excessive worries, and appropriate treatments were initiated to treat these problems with the potential of mitigating the severity and impact of fatigue. Thus, the rehabilitation program comprised a combination of treatments for interrelated problems. The EM model is only concerned with self-management of fatigue, and any treatment options for reducing fatigue need to be pursued before (or simultaneously with) initiating EM. Distinguishing the targets and aims of these complex and intertwined processes in rehabilitation may promote deliberate tailoring of services.

### Study Limitations

First, the model is based on one rehabilitation center used as a case of clinical practice. Thus, any diverging practice in other settings may not be represented by the model at this stage. The center is a large and specialized rehabilitation center with a long history of providing rehabilitation services. Further, several of the components revealed in this study are consistent with previous research in fatigue management in ABI populations, e.g., activity pacing, intermittent breaks, fatigue/activity monitoring, and knowledge and acceptance of fatigue [3,16,17,50]. The model is not considered finite, and further research in other clinical settings is encouraged to refine propositions of the model. In addition, while the model maps targets and ingredients of EM, data was inadequate to specify the hypothesized mechanisms of action, as participants had difficulties articulating mechanisms and distinguishing this element from the others. Addressing how treatments are expected to work is crucial to clinical reasoning, and defining and testing mechanisms of action is an important next step of this model.

Second, two eligible individuals declined to participate due to fatigue and another was withdrawn, which limits generalizability of results. Further, individuals with other types of ABI than stroke were not represented in the case study. The impact of fatigue need not be disease-specific, however, and elements of EM are used across several long-term conditions such as multiple sclerosis [51] and cancer [52]. The extent to which management of fatigue generalizes across different types of ABI and other long-term conditions is an avenue for future research [53,54,55].

Third, although the RTSS-based log was administered with instructions and an illustrative example, several log entries were not compatible with RTSS. Interviews provided high-quality data due to the interactive nature of this method, but they were only conducted after rehabilitation to examine EM in retrospect. Despite these limitations, the use of multiple methodologies (co-production workshops and case study design), data sources (service providers and users), and data collection methods in a longitudinal design (interviews, logs, and questionnaires) enabled triangulation to provide a comprehensive account of EM.

## 5. Conclusions

Informed by routine practice in a neurorehabilitation setting, a treatment model for management of fatigue was specified using RTSS as a conceptual framework. This model describes contents and theories of a set of treatment components, which are concerned with promoting self-management of fatigue symptoms in daily living after ABI. Individual treatment components are framed according to their hypothesized active ingredients, and the model distinguishes between different treatments in the management of fatigue. Thus, the EM model may contribute to delineating complex and intertwined processes of person-centered rehabilitation programs by guiding the clinician in selecting and adjusting appropriate treatments. Further, the EM model may facilitate theory-driven evaluation research by outlining a set of hypothesized treatment components. Consequently, the model may facilitate research on a complex and widespread approach in routine practice currently lacking a robust evidence base.

## Figures and Tables

**Figure 1 jcm-12-03192-f001:**
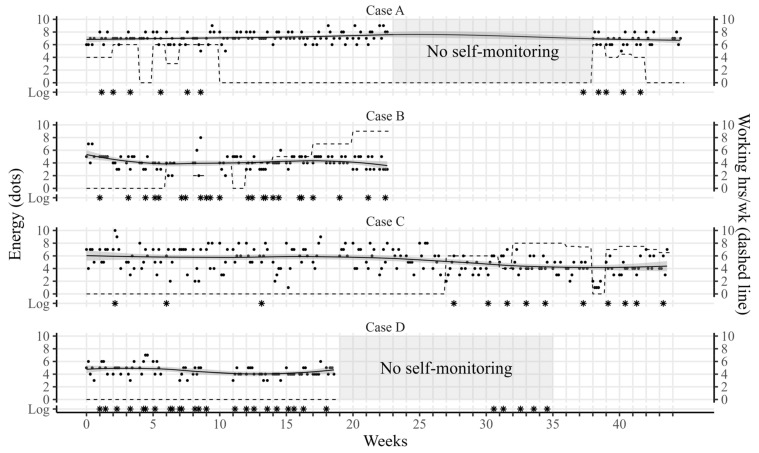
Timelines of rehabilitation programs by case. Note: Day-to-day energy self-monitoring is represented by dots and summarized using locally estimated scatterplot smoothing (solid line). Dashed lines represent self-reported working h/wk. Asterisks represent individual treatment sessions reported in log entries.

**Table 1 jcm-12-03192-t001:** Coding protocol.

First-Order Level Category	Description
Ingredients	Observable (and, therefore, in principle, measurable) actions, words, hands-on manipulation, common objects, chemicals, devices, or forms of energy that are selected/delivered by the clinician to a treatment recipient. *
Context	Conditions of the physical and social environment and personal characteristics of the treatment recipient that affect mechanisms of action or the choice of ingredients.
Actors	Stakeholders (individuals, groups, and institutions) who play a role in the rehabilitation program.
Mechanism of action	Process by which a treatment’s active ingredients induce change in the target of treatment. *
Target	Specific, measurable (in principle) aspect of the recipient’s functioning or personal factor that is predicted in the treatment theory to be directly changed by the treatment’s mechanism of action. *
Aims	Aspect(s) of the patient’s or other recipient’s functioning or modifiable personal factors that may or may not change indirectly (via mechanisms specified in enablement theory) as a result of the treatment-induced change in the treatment target or in multiple treatment targets. *

* Description is quoted from The Manual for Rehabilitation Treatment Specification V. 6.2 [22] (pp. 60–64).

**Table 2 jcm-12-03192-t002:** Case characteristics at study inclusion.

	Case			
Variable	A	B	C	D
Sex	Male	Male	Female	Female
Age	59	60	62	57
Educational level	Bachelor’s degree	High school	Vocational	Bachelor’s degree
Civil status	Cohabiting	Single	Single	Married
Children (residing|non-residing)	0|3	1|2	0|0	0|3
Injury type	Ischemia	Hemorrhage	Ischemia	Ischemia
Lesion lateralization	Right	Right	Left	Right
Months since injury	10	7	12	10
Previous brain injury	No	No	No	Yes
Pre-injury fatigue	Yes, but different	No	Yes, but different	No
Habitual working h/wk	37	37	32	37
Financial basis	Sickness benefits	Sickness benefits	Wages	Sickness benefits
Employment status				
Pre-injury	Unemployed	Employed	Employed	Unemployed
At inclusion	Unemployed	Unemployed	Sick leave	Unemployed
Log entries, no. (h)				
Occupational therapist	11 (10.67)	18 (14.83)	13 (13.00)	18 (15.50)
Neuropsychologist	0 (0.00)	3 (4.00)	0 (0.00)	10 (9.33)
Physiotherapist	0 (0.00)	3 (2.50)	0 (0.00)	0 (0.00)
Total	11 (10.67)	24 (21.33)	13 (13.00)	28 (24.83)

**Table 3 jcm-12-03192-t003:** The Energy Management model: Treatment components for self-management of fatigue.

Target	Group	Ingredient
KNOWLEDGE AND UNDERSTANDING OF FATIGUE		
Enhanced knowledge and understanding of fatigue: The concept of fatigue as a consequence of injury Interactions between fatigue symptoms and activities Triggers of fatigue symptoms Signs and symptoms of fatigue Limitations posed by fatigue Needs due to fatigue in daily life	R	Provide semantic information on fatigue (verbally or using visual models/analogies) Discuss how information applies to experiences; Share peer experiences; Discuss responses on formal assessment Fatigue/activity diary (in any preferred format) Provide rationale and instructions for use and tailor the format to individual needs; Provide opportunities for use at home; Troubleshoot any barriers Guided analysis of diary entries Probe reasons for fluctuations in energy levels; Query effects of various activities, life circumstances, and changes in daily routines; Categorize activities by effect Query early signs of fatigue in recent situations
INTEROCEPTIVE ATTENTION OF FATIGUE		
Increased ability to notice signs of fatigue Form a habit of responding to signs of fatigue	S	Guided use of a 10-point energy scale Query sensations on different levels of the scale; Encourage and guide use at home Fatigue/activity diary (see Knowledge and Understanding of Fatigue) Guided practice in mindfulness techniques Provide rationale and instructions; Provide opportunities to practice at home Discuss reactions to symptoms of fatigue and management options Direct attention to overt signs of fatigue and guide use of management strategies
ACCEPTANCE OF FATIGUE		
Increased recognition of fatigue as a chronic condition Adapted expectations to current functional level	R	Acknowledge the impact of fatigue Document fatigue by formal assessment; Discuss limitations posed by fatigue Discuss expectations regarding recovery/persistence of fatigue Discuss current vs. previous functional level Challenge perspectives on life roles, core values, and identity Opportunities for sharing peer experiences
ACTIVITY MANAGEMENT		
Form a habit of managing fatigue in daily activities: Scheduling activities Planning according to anticipated energy levels Alternating periods of activity/rest Reappraising activities Managing tasks efficiently	S	Week planner, daily schedule, etc. Encourage regular use Guided practice in planning activities based on energy conservation strategies: Distribute exerting activities; Plan rest and recovery ahead of exertion; Plan regular rests/breaks; Adjust (and grade) overall activity level Provide opportunities to practice planning at home Provide rationale and instructions; Troubleshoot any barriers for using planning strategies; Evaluate effects with recipient Encourage intermittent breaks and activity pacing Probe valued activities, duties, and responsibilities Encourage prioritizations and guide if needed; Encourage redelegating tasks or suggest alternative solutions to managing duties; Troubleshoot barriers for engaging in valued activities Encourage strategies to reduce physical and cognitive load in daily task management
Increased ability to rest efficiently	S	Provide opportunities to test relaxation techniques and types of rest at home Query preferences and provide information; Troubleshoot any barriers; Evaluate effects with recipient
SELF-MANAGEMENT OF FATIGUE		
Increased confidence in one’s ability to manage fatigue in daily living	R	Graded support/independence in implementation of strategies as needed Evaluate effects of strategies with recipient Query options for managing fatigue in challenging situations in daily life

Note: The treatment model was specified using the Rehabilitation Treatment Specification System. All targets are direct targets involving recipient volition. S = Skills and Habits, R = Representations.

## Data Availability

Data are unavailable due to privacy.

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
