# Peer review of "Defining a Treatment Model for Self-Management of Fatigue in Rehabilitation of Acquired Brain Injury Using the Rehabilitation Treatment Specification System"

_jcm, 2023, doi:10.3390/jcm12093192_

Round 1

Reviewer 1 Report

Thank you for this well-written manuscript describing your novel project. It is exciting to see this application of the RTSS. This project demonstrates its potential value in understanding and applying treatment options. I offer the following suggestions:

1. It would be helpful to readers to offer an explanation of the meaning of a "treatment component" since the term is used several times in the manuscript.  One possibility could be to revise the statement on lines 47-49 to read: 

In this framework, a "treatment component" comprises three elements: Ingredients (what the clinician does), Mechanism of Action (how the treatment is expected to work), and a Target (the aspect of functioning hypothesized to be directly changed by the applied ingredients).  

2. p. 3, lines 106-10:

- What training or orientation was provided to providers to help them understand how to apply RTSS terminology?

- It is mentioned that the providers logged up to three "pairs" of ingredients, mechanisms of action, and targets?  Could multiple ingredients be listed as part of that pair? How likely is it that providers might have delivered more "pairs" but could not document them given the limitation?

3. Table 1: This may be to subtle a point, but I recommend stating the first level category for mechanism of action and target in singular form to reinforce that each treatment component has only one target and one mechanism.

4. I find the center alignment of the text in Table 3 difficult to read and request it be changed to left-aligned if journal guidelines allow.

5. Table 3, p. 6, "Increased ability to rest efficiently row": The ingredients listed seem more like R group ingredients focused on awareness of preferences and self-evaluation. Opportunities to practice are an essential ingredient of S group interventions, but are not listed. I wonder if the investigators might consider re-framing the target as "Increased awareness of strategies to facilitate efficient rest" with the actual rest considered as a aim (a downstream effect of the improved awareness).

6. I noted that while providers were asked to note the mechanism of action associated with each target and associated ingredients, mechanisms of action were not discussed in the findings. Is this because providers were uncertain about the mechanisms or because there were discrepancies in how mechanisms were stated among clinicians, or for other reasons?  Understanding any difficulties encountered in defining mechanisms would be very helpful to the designers and users of the RTSS, as these experiences may suggest a need for further guidance on the approach to articulating the mechanism of action portion of treatment theory. Please consider addressing the specification of mechanisms explicitly, as part of the discussion and/or results sections.

Thank you for the opportunity to review this manuscript.

Author Response

Thank you for your valuable review and suggestions for improving the manuscript. Please find our response to your comments and suggestions below.

Point 1: It would be helpful to readers to offer an explanation of the meaning of a "treatment component" since the term is used several times in the manuscript.  One possibility could be to revise the statement on lines 47-49 to read:

In this framework, a "treatment component" comprises three elements: Ingredients (what the clinician does), Mechanism of Action (how the treatment is expected to work), and a Target (the aspect of functioning hypothesized to be directly changed by the applied ingredients).

Response 1: Thank you for this helpful suggestion. We have revised the manuscript accordingly.

Point 2: p. 3, lines 106-10:

- What training or orientation was provided to providers to help them understand how to apply RTSS terminology?

- It is mentioned that the providers logged up to three "pairs" of ingredients, mechanisms of action, and targets?  Could multiple ingredients be listed as part of that pair? How likely is it that providers might have delivered more "pairs" but could not document them given the limitation?

Response 2: Regretfully, training was limited, and several log entries were not that informative in relation to RTSS, as noted in study limitations on page 9, Lines 314-315. Service providers received instructions to the treatment log and an illustrative example of a fictional log entry, but more training could have been helpful to ensure better application of RTSS terminology. We have added the following information to the manuscript, p. 3, Lines 121-122:

“The log was pilot tested and revised prior to the case study, and service providers received instructions to the log and an illustrative example.”

Multiple ingredients could be listed as part of one “pair” / component, as responses were recorded in free text fields without word count limitations. Thus, providers were free to elaborate. It is possible that some pairs were not documented given the limitation on three pairs. Service providers were instructed to report on the three most important activities in the case they were doing several things in that session. Often, however, less than three pairs were reported. Service providers were managing several problems during the rehabilitation program, including liaising during work trials in vocational rehabilitation, and they were instructed only to report on treatment activities related to fatigue management.

Point 3: Table 1: This may be to subtle a point, but I recommend stating the first level category for mechanism of action and target in singular form to reinforce that each treatment component has only one target and one mechanism.

Response 3: You are completely right, thank you for pointing that out. We have revised the text as suggested.

Point 4: I find the center alignment of the text in Table 3 difficult to read and request it be changed to left-aligned if journal guidelines allow.

Response 4: Please see Response 5.

Point 5: Table 3, p. 6, "Increased ability to rest efficiently row": The ingredients listed seem more like R group ingredients focused on awareness of preferences and self-evaluation. Opportunities to practice are an essential ingredient of S group interventions, but are not listed. I wonder if the investigators might consider re-framing the target as "Increased awareness of strategies to facilitate efficient rest" with the actual rest considered as a aim (a downstream effect of the improved awareness).

Response 5: Thank you very much for noticing this concern. I am afraid the text alignment was altered after submission, which causes confusion on the relationships between targets and ingredients. In fact, the ingredient above, “Provide opportunities to test relaxation techniques and types of rest at home”, is also related to this target, and the other ingredients were listed as part of this ingredient. However, the target has been center-aligned vertically, which moved the target one line below this ingredient. Thus, as you mention, we intended to list an “opportunity to practice”-ingredient to this S group target.

Based on data analysis, we figured that the target should be framed as an S group intervention. We found out that service providers were providing opportunities to practice relaxation techniques to improve the ability to rest. They were also providing information about strategies to facilitate rest and queried individual preferences (as listed in the ingredients), but this was in the process of selecting strategies for practice that might enable the individual to rest more efficiently (S target). Thus, we believe that the S group specification is correct, and that this confusion is due to the misalignment of the text, which erroneously moved a very central ingredient outside the scope of the target. We will make sure to fix the alignment of targets and ingredients. Treatment specification is not straightforward, however, so please let us know if you still think the target needs to be re-specified? One may also consider including a separate R target, like the one you mention, but service providers were only providing information about relaxation techniques to practice and evaluate the use of these strategies.

Point 6: I noted that while providers were asked to note the mechanism of action associated with each target and associated ingredients, mechanisms of action were not discussed in the findings. Is this because providers were uncertain about the mechanisms or because there were discrepancies in how mechanisms were stated among clinicians, or for other reasons?  Understanding any difficulties encountered in defining mechanisms would be very helpful to the designers and users of the RTSS, as these experiences may suggest a need for further guidance on the approach to articulating the mechanism of action portion of treatment theory. Please consider addressing the specification of mechanisms explicitly, as part of the discussion and/or results sections. 

Response 6: Thank you for this comment. We agree with you on this matter, and we believe that specifying the mechanisms of action is an important next step for this model. Unfortunately, we faced difficulties defining the mechanisms based on the data available, as study participants had trouble articulating this element, and I guess we as investigators lacked tools for addressing this topic adequately. Thus, we left this portion out and focused on mapping the ingredients and targets of EM. We have addressed this concern in the discussion section, p. 9, Lines 309-314:

“In addition, while the model maps targets and ingredients of EM, data was inadequate to specify the hypothesized mechanisms of action, as participants had difficulties articulating mechanisms and distinguishing this element from the others. Addressing how treatments are expected to work is crucial to clinical reasoning, and defining and testing mechanisms of action is an important next step of this model.”

Reviewer 2 Report

I think the concept behind this paper is very interesting, but there are aspects of the paper that could be improved. Comments will be provided under each manuscript section heading.

1. Introduction:

- I thought overall the Introduction was well written and provides a concise and clear background to the study. More detail could be provided re the origin of the term 'Energy Management' and the domains listed as part of this - is this an established term in fatigue management after ABI, or was it derived from the 2 publications cited?

2. Materials and methods:

- This section does not present a coherent explanation of the study method and needs considerable revision. For example, it is not clear how the activities listed under each phase contributed to and were incorporated into the developed treatment model. 

- Phase 1 is fairly well described and the outcome of a draft model of treatment seems clear. However perhaps more detail re the initial treatment theories that were specified could be useful.

- The purpose and contribution of Phase 2 to the development of the draft model of treatment is not explained.

- Why was a realist approach used for the qualitative interviews and what did this entail? A copy of the interview guides would be useful.

- Line 110, p.3 states there were 8 interview transcripts but it is not clear who these were involved - in the Results section, line 137 p. 4 says that four participants completed the study. Assuming that these were service-users, were the other 4 transcripts from interviews with service providers? It appears from Table S1 that only 2 service providers were interviewed. 

- It is not clear what the EM model discussed in section 3.1 Treatment model on EM is - is this the model that was developed in Phase 1 with service providers? It has not previously been called this. What was the process for incorporating the Phase 2 outcomes into this? 

- Table 3 is hard to understand due to the way it is set out. It is also not clear how all the detail in this table was derived and how the qualitative data subsequently reported in section 3.1.1-3.1.5 contributed to this.

- I am unclear as to what contribution the data in Figure 1 made to the development of the model.

Author Response

Thank you very much for your valuable comments to improve the manuscript. Please find our response to your comments and suggestions below.

Point 1: 1. Introduction: I thought overall the Introduction was well written and provides a concise and clear background to the study. More detail could be provided re the origin of the term 'Energy Management' and the domains listed as part of this - is this an established term in fatigue management after ABI, or was it derived from the 2 publications cited?

Response 1: Thank you. The term “Energy Management” is frequently used among practitioners in Denmark (in Danish, “energiforvaltning”) to describe a common treatment approach for fatigue management. This approach is much alike the one described in the two publications cited. Thus, “Energy Management” is translated from a term used in Danish. We have elaborated on the origin of the term in the revised manuscript, p. 1, Lines 36-42:

“In Denmark, this approach is commonly termed Energy Management (EM) and is widely used in clinical practice to reduce the burden of fatigue. It comprises various educational, psychological, and behavioral strategies such as psychoeducation, symptom monitoring, and planning of activities and rest. However, the specific content and the hypothesized active ingredients of treatment are poorly defined, and clinicians rely predominantly on their own knowledge and experience when managing fatigue [17–19].”

Point 2: 2. Materials and methods: This section does not present a coherent explanation of the study method and needs considerable revision. For example, it is not clear how the activities listed under each phase contributed to and were incorporated into the developed treatment model.

Response 2: We have revised and restructured the Materials and methods section substantially. To provide the reader with a coherent overview of the method, we have revised and extended the lead-in description of the study design, including how each phase contributed to the development of the treatment model, p. 2, Lines 64-70:

“A multi-phase qualitative study was conducted to specify a treatment model on EM based on investigations of clinical practice in management of fatigue. Phase 1 entailed co-production workshops with practitioners to formulate an initial draft of the treatment model. In phase 2, a collective case study was conducted to examine how treatment processes unfold in clinical practice. Based on case analysis, the initial draft was refined, finalized, and reported as the EM model.”

In addition, we have added a subsection at the end, 2.3. Model specification, explaining how results from phases 1 and 2 were incorporated into the development of the final treatment model, p. 4, Lines 147-153.

2.3. Model specification

The treatment components identified in case analysis in phase 2 were compared with the components of the initial draft of the model derived from phase 1. The final organization of treatment components and the targets and ingredients of each component were re-specified based on results from both phases using The Manual for Rehabilitation Treatment Specification [22]. Targets and ingredients identified in both phases were merged and incorporated into the final treatment model, the EM model.”

Point 3: Phase 1 is fairly well described and the outcome of a draft model of treatment seems clear. However perhaps more detail re the initial treatment theories that were specified could be useful.

Response 3: We have deliberately omitted the results of the preliminary work in phase 1 to avoid confusing the reader about the final outcome of the study. However, we recognize the need to mention the outcome of phase 1 in brief, at least. Thus, we have added a brief description of the draft model and clarified to the reader, p. 2, Lines 91-94:

“Initial treatment theories included detailed ingredients related to, e.g., knowledge, understanding, and acceptance of fatigue, and planning, prioritizing, and adapting activities. The results of this preliminary work in phase 1 were incorporated into the final EM model.”

Point 4: The purpose and contribution of Phase 2 to the development of the draft model of treatment is not explained.

Response 4: We have reorganized the description of Phase 2, and the contribution of Phase 2 to the development of the treatment model is now described in more detail in the new subsection, 2.3. Model specification (please see Response 2). Further, the contribution of Phase 2 is also mentioned more explicitly in the lead-in to the Materials and methods section (please see Reponse 2).

Point 5: Why was a realist approach used for the qualitative interviews and what did this entail? A copy of the interview guides would be useful.

Response 5: The realist approach is based upon a realist philosophy of science, which is very useful for studying and understanding causal effects in complex interventions such as a rehabilitation program. The realist approach (please see reference 28 in the revised manuscript) to evaluation research is largely theory-driven and is concerned with exploring and testing propositions about how, where, when, and why treatments work through the views and experiences of participants. As we wanted to specify treatment theories based on practice-based knowledge, understandings, and routines, we figured that the realist approach would be helpful in investigating stakeholders’ perspectives on treatment theories of energy management.

I am afraid we are not able to provide a helpful copy of the complete interview guide, as it is in Danish only. However, we will share the materials upon request from readers. If reviewers prefer, we can translate main themes in the interview guide if needed. Further, the interview guide was based on materials from the RAMESES II project, which is cited in the manuscript. We also cite Pawson and Tilley (reference 28), who provide a thorough introduction to realist evaluation. We would argue that it is outside the scope of this manuscript to provide a detailed explanation of the realist approach, and the interested reader is advised to consult these cited resources.

Point 6: Line 110, p.3 states there were 8 interview transcripts but it is not clear who these were involved - in the Results section, line 137 p. 4 says that four participants completed the study. Assuming that these were service-users, were the other 4 transcripts from interviews with service providers? It appears from Table S1 that only 2 service providers were interviewed. 

Response 6: Interviews were conducted with service users and service providers, respectively. Thus, two interviews per case were conducted. As two of the service providers were involved in two cases each, both were interviewed twice. We have rephrased Line 111-113 (p. 3) to clarify:

“For each case, two interviews were conducted by author F.D. one-on-one with the service user (ca. 45–75 min.) and a service provider (ca. 45–60 min), respectively, at the end of rehabilitation using a realist approach [28].”

Point 7: It is not clear what the EM model discussed in section 3.1 Treatment model on EM is - is this the model that was developed in Phase 1 with service providers? It has not previously been called this. What was the process for incorporating the Phase 2 outcomes into this? 

Response 7: Thank you for bringing this cause of confusion to our attention. While restructuring the Materials and methods section (please see Response 2), we have made sure to introduce the term “EM model”, which is the name of the final treatment model that was developed throughout Phase 1 and 2. The contribution of Phase 1 and 2 to the treatment model is also described in more detail (please see Response 2).

Point 8: Table 3 is hard to understand due to the way it is set out. It is also not clear how all the detail in this table was derived and how the qualitative data subsequently reported in section 3.1.1-3.1.5 contributed to this. 

Response 8: Unfortunately, the layout of the table has been altered after submission, making it difficult to read. We will check with the editor if it is possible to correct the layout.

The intended format follows the conventions in the Manual for Rehabilitation Treatment Specification. The table presents the final treatment model that was derived from Phase 1 and 2. The new subsection, 2.3. Model specification, elaborates on how these phases - and the qualitative data - contributed to the targets and ingredients reported in the model (please see Response 2). The table presents the five treatment components of the model, and each section from 3.1.1-3.1.5. explains the contents of each component and links to the qualitative data derived from the case study (Phase 2) to support the findings.

Point 9: I am unclear as to what contribution the data in Figure 1 made to the development of the model. 

Response 9: The model was developed, in part, based on the case study, and Figure 1 serves to characterize the specific cases that were studied. Thus, much like Table 2, the figure hopefully provides the reader with detailed information about the cases and their rehabilitation trajectories, which may help contextualize the findings and support transferability across settings and various types of rehabilitation programs as the reader interpret and apply the findings in his/her own setting. Thus, we believe Figure 1 is useful for the interpretation of the findings, but we are willing to move this figure to the supplemental materials, if reviewers think it distorts the ease of reading.

Reviewer 3 Report

Journal of Clinical Medicine

Manuscript ID: jcm-2253659

Title: Defining a treatment model for self-management of fatigue in rehabilitation of acquired brain injury using the Rehabilitation Treatment Specification System

Authors: Dornonville de la Cour, Norup, Andersen, and Schow.

Thank you for the opportunity to review this manuscript.

The authors of this study conducted a qualitative multi-phasic study in Denmark to formulate a treatment model to manage fatigue in individuals with ABI. The phases of their study included workshops with service providers and case studies. The final treatment model utilizes the Rehabilitation Treatment Specification System framework which focuses on ingredients in treatment (what the clinician does), mechanisms of action (how the treatment is expected to work), and targets (aspects of functioning directly targeted for change). Using this framework helped authors address the issue of the lack of specificity around treatments in TBI, particularly to treat fatigue. The authors devised a treatment that is comprised of five key components that can serve as the basis for this much- needed treatment in ABI.

Overall the paper highlights a needed and methodological sound qualitative study that outlines an intervention to help patients with ABI manage fatigue. It sets the stage for the future work of these authors and others in the field. The following few suggestions are provided for the authors’ consideration:

Key points

The qualitative study design was well thought-out and the study goals well-conceptualized. The authors were thoughtful in their inclusion/exclusion criteria of patients.

The detailed ingredients, mechanisms of action, and targets are very helpful and give the audience a solid understanding of the nature of the intervention.

Authors adequately addressed the study limitations including limited generalizability to all ABI populations, including those with severe fatigue (given 2 participants were excluded / declined due to fatigue.

Minor points

It is not clear why authors chose 3-24 months for exclusionary criteria and how large the potential pool of ‘included’ patients. Please include data from this Danish health care system regarding the number of patients served per year who are 18-65 with an ABI. It seems like only 10 people screened in 13 months is low. It is also surprising there are no patients with TBI included. Were any of those excluded presenting with a history of this etiology?

Author Response

Thank you very much for the thorough review of the manuscript and the valuable comments. Please find our response below.

Point 1: It is not clear why authors chose 3-24 months for exclusionary criteria and how large the potential pool of ‘included’ patients. Please include data from this Danish health care system regarding the number of patients served per year who are 18-65 with an ABI. It seems like only 10 people screened in 13 months is low. It is also surprising there are no patients with TBI included. Were any of those excluded presenting with a history of this etiology?

Response 1: The criterium regarding time since injury were chosen to recruit patients having had sufficient time after injury to experience problems with persistent fatigue in daily life, but not for long enough for them to establish unhelpful habits and routines in fatigue management that may be difficult or time consuming to change and adapt in treatment.

We have added information from the Danish Health Data Authority on the number of patients with ABI per year, p. 3, Lines 107-109:

“In Denmark, 6.102 adults between 18–64 years old are hospitalized with an ABI per year [27]. However, it is unclear how many of these need vocational rehabilitation after hospital discharge.”

Recruitment was indeed an issue for several reasons. First, we faced two covid-19 lockdowns in Denmark, which delayed the rehabilitation programs, and we received less referrals for vocational rehabilitation during lockdowns. Second, in the past years, Danish municipalities are increasingly providing rehabilitation services themselves instead of referring people to specialized brain injury centers. Thus, while the study was going on, we received fewer patients with ABI than expected and more tasks related to supervision and counselling. Therefore, we only screened 10 people with ABI for study enrollment. Unfortunately, I do not have a list with details on those excluded due to change of affiliation and data privacy, so I cannot tell if any of those excluded presented with a history of TBI. Nevertheless, generalization of results is limited, as the cases only included people with stroke, although both stroke and TBI (and other types of ABI) were considered in the co-production workshops in phase 1.